# OpenReview forum: "Teacher-Student Multi-Agent Reinforcement Learning Framework for AutoML Pipeline Construction"
_ICLR.cc/2026/Conference — ICLR 2026 Conference Withdrawn Submission_

### Official Review · Reviewer_Vqqa · 2025-10-28

**Soundness:** 1
**Presentation:** 1
**Contribution:** 2
**Rating:** 2
**Confidence:** 4

**Summary:**

This paper proposes a framework for AutoML pipeline construction using an asymmetric teacher-student multi-agent reinforcement learning (MARL) approach. The core idea is to frame pipeline synthesis as a Decentralized Partially Observable Markov Decision Process (Dec-POMDP) where a "student" agent proposes pipeline components and a "teacher" agent selectively intervenes. The authors claim their method matches or surpasses baselines in final accuracy while requiring 35-60% fewer evaluations, and that the framework is more interpretable and supports knowledge transfer.

**Strengths:**

1.  The pedagogical motivation of a teacher guiding a student provides an intuitive inductive bias for the AutoML search process.
2.  The paper aims to address a critical bottleneck in AutoML—the high cost of pipeline evaluation.

**Weaknesses:**

1.  The authors explicitly state in Section 5.8 that the quantitative ablation table was omitted because prior results became invalid after code refactoring. This is a critical omission for a paper whose main contributions are new algorithmic components.
2.  The description of the framework in Section 3 lacks clarity. Key concepts, particularly the Dec-POMDP formulation, are introduced without sufficient definition. The mapping between the formal elements of the Dec-POMDP (states, observations, transition functions) and the concrete task of building a machine learning pipeline is not well-explained, making the method difficult to understand.
3.  The main performance results in Table 1 show that the proposed MARL approach only marginally outperforms the strongest baseline (H2O AutoML) on most datasets.
4.  The related work section is quite brief and could be expanded to better contextualize the work. Furthermore, figures are difficult to read.

**Questions:**

1. The omission of the ablation study is a critical issue. Can you provide results from ablation studies?
2. Could you please provide a clearer explanation of your method?

---

### Official Review · Reviewer_KcKR · 2025-10-31

**Soundness:** 1
**Presentation:** 1
**Contribution:** 1
**Rating:** 0
**Confidence:** 3

**Summary:**

The authors present an asymmetric teacher–student multi-agent reinforcement learning framework for automated machine learning and compare it against baselines like Random/Grid Search, single-agent DDQL, TPOT.

**Strengths:**

None. The motivation behind the paper's methodology is lackluster, the problem statement is unclear, and more; see the weaknesses.

**Weaknesses:**

I cannot in good faith recommend this paper for acceptance, and I strongly recommend rejection. Without even reading the disclosure of LLM usage at the end of the article, I already knew it was almost entirely LLM-generated. The writing is incoherent, disorganized, poorly motivated, and abrasive in tone.

Abstract:
* More motivation is required. Too much focus on your method without talking about the big picture. Hence, your problem statement is unclear. What are the challenges of AutoML? You mention monolithic search, but I doubt that there is no literature addressing this.
* Strong baselines? Random/Grid Search and single-agent DDQL are not strong baselines. And DDQL should be spelled out - is it double deep q learning or dueling deep q learning?

Introduction:
* I find the wording very confusing: teacher-student, curriculum pacing, etc. You should relate this more closely to asymmetric pedagogical control. Also, asymmetrical with respect to what?
* The presentation is a disorganized mess. The problem statement is barely established. There are bold claims (e.g., "attains competitive or superior accuracy", "reduces evaluations 35-60%", "yields interpretable credit traces", "transfers knowledge with up to 40% episode reduction"); none of this sounds scientific or statistical. Where are the means, standard deviations, confidence intervals, etc.? These percentages could be outliers.
* Where are your citations for "Positioning vs LLM Orchestration"? That entire section needs evidence.

Related Work:
* NAS: you need to spell this out = Neural Architecture Search (NAS)
* Too short and brief; seems incoherent.

Framework:
* 3.2: This does not seem novel at all if this is  ASYMMETRIC DEC-POMDP. Also, what does DEC-POMDP stand for?

Results:
* Poor selection of visualizations for displaying results. Performance looks bad and unstable. Unprofessional y-axis labels for Figure 1, bottom right plot. Ridiculous and unnecessary scatter plot in Figure 2. Confusing Figure 3 with a shared y-axis between unrelated metrics. Deceptive Figure 4 dual axes plot with bar plot and line plot.

5.1: "Auto-sklearn removed due to instability". I feel that this needs further discussion or investigation.
5.2: Incoherent and lackluster results.
Line 269: There is an upside-down question mark that should not be there
Line 313: put a period after your sentences

This submission does not meet ICLR quality standards due to its lack of clear presentation or convincing/rigorous results.

**Questions:**

See Weaknesses.

**Details Of Ethics Concerns:**

I have worked with ChatGPT enough to recognize its output and tone. Almost the entirety of this submission is written by ChatGPT, and it should not, by any means, be considered for acceptance to ICLR. Regardless of how it was authored, the paper is incomplete, incoherent, and disorganized.

---

### Official Review · Reviewer_Bim1 · 2025-10-31

**Soundness:** 2
**Presentation:** 3
**Contribution:** 3
**Rating:** 4
**Confidence:** 3

**Summary:**

This paper introduces an asymmetric teacher–student multi-agent RL (MARL) framework for AutoML pipeline synthesis. The approach frames AutoML pipeline construction as a Decentralized POMDP, in which a student agent sequentially builds pipeline components, and a teacher agent selectively intervenes only when the expected counterfactual advantage exceeds a dynamic threshold. Through this selective pedagogical intervention, the system accelerates early learning while fostering autonomy later, mimicking human teaching dynamics. A component-level credit assignment mechanism estimates marginal contributions using sparse ablations and historical reuse, yielding interpretable attributions and enabling transfer across datasets. Empirical results on several tabular benchmarks (Iris, Adult, Covertype, Credit-G, Bank Marketing) demonstrate good performance.

**Strengths:**

The formulation of AutoML pipeline construction as an asymmetric MARL problem is interesting. Framing the teacher as a counterfactual intervention policy introduces a pedagogical inductive bias that aligns with human-guided learning metaphors.

**Weaknesses:**

1) The ablation section is largely qualitative, citing environment refactors as justification for omission of quantitative deltas. This prevents precise attribution of performance gains to specific modules (teacher gating, adaptive exploration, or credit approximation).

2) Some key hyperparameters are manually chosen. The lack of sensitivity analysis limits claims of generality and robustness.

3) Evaluation is limited to small and mid-scale tabular datasets. There is no evidence of scalability to large-scale, high-dimensional, or multi-modal AutoML tasks.

**Questions:**

Q1: The paper reports minor negative transfer (e.g., Iris -> Adult). Can the authors isolate whether this originates from the teacher’s overfitting or student policy misalignment? During transfer, does freezing the teacher always outperform joint fine-tuning in stability?

Q2: The decay is heuristic. How sensitive are the results to the chosen decay parameters?

Q3: The system produces structured artifacts (credit weights, intervention logs). Could these be visualized or summarized in a way that aids human debugging? Is there any qualitative user study or visual audit confirming the interpretability claims?

---

### Official Review · Reviewer_GEG1 · 2025-10-31

**Soundness:** 2
**Presentation:** 1
**Contribution:** 2
**Rating:** 2
**Confidence:** 4

**Summary:**

The paper proposes a asymmetric teacher-student multi agent framework for automated machine learning pipeline synthesis in which the teacher can propose a counterfactual improvement based on the performance of the student agents. They propose component-level credit assignment solution with policy warm start to improve the performance of student learner. The author also test their methods on various dataset and have showed superior performance compared to several baselines.

**Strengths:**

1. The paper proposes an interesting method for AutoML decision by framing it as a teacher-student model where teacher uses a advantage estimator to guide the student towards correct solution.

2. The authors have conducted detailed experiments and have provided good comparison against various baseline algorithms.

**Weaknesses:**

1. The paper is poorly written which obscures the technical content of the paper. The authors have chosen a style with lot of declarative sentences making the text feel fragmented and difficult to follow the narration.
2. Related works section lacks a proper comparison of their work with prior literature.
3. Definitions of lot of important terms like component pipeline, many terms like component library, valid ordered pipeline, utility function etc are not well defined. State and observation space definition, rewards, transition - these all definitions are missing in Dec-POMDP. What should one even understand from this in line 103 “Transition integrates component attachment or termination with evaluation (cross-validation + timeout guard).”?

**Questions:**

1. Can you please clarify the problem setting and Dec-POMPD framework to be more understandable to reader?
2. Can you clarify what component library means?
3. A lot of wordings in the text are not at all clear to me as a reader. For example, can you clarify what do you mean by the text "Credit-weighted marginal gain detection suppresses gratuitous component accretion, yielding emergent complexity pruning without explicit length penalty escalation." in line 256-257?

---

### Official Review · Reviewer_RAAx · 2025-11-01

**Soundness:** 1
**Presentation:** 1
**Contribution:** 1
**Rating:** 0
**Confidence:** 3

**Summary:**

This paper presents a novel asymmetric teacher–student multi-agent reinforcement learning (MARL) framework for automated machine learning (AutoML) pipeline synthesis, which uses a selective counterfactual intervention mechanism for efficient guidance.

**Strengths:**

I think this paper is still unfinished. For example, one section is marked as preliminary. Therefore, it is difficult to fully assess the quality of the work at this stage.

**Weaknesses:**

The paper is poorly organized, with too many subsections. Sentences and paragraphs lack coherence. The references are insufficient, and the experimental results are presented in a rather casual manner. Overall, it feels more like a draft manuscript than a polished paper ready for publication.

**Questions:**

I suggest that the authors further improve the paper by adding more references and experimental results, and by reorganizing the paragraphs. The current version appears more like an AI-generated preliminary draft.

---

### Note · Authors · 2025-11-27

I have read and agree with the venue's withdrawal policy on behalf of myself and my co-authors.